

# Evaluating the performance of a Picarro G2207-*i* analyser for high-precision atmospheric O₂ measurements

Leigh S. Fleming[1], Andrew C. Manning[1], Penelope A. Pickers[1], Grant L. Forster[1,2], and Alex J. Etchells[1]

[1]Centre for Ocean and Atmospheric Sciences, School of Environmental Sciences, University of East Anglia, Norwich, UK
[2]National Centre for Atmospheric Science, University of East Anglia, UK

*Correspondence to*: Leigh Fleming (leigh.fleming@uea.ac.uk)

**Abstract.**

Fluxes of oxygen ($O_2$) and carbon dioxide ($CO_2$) in and out of the atmosphere are strongly coupled for terrestrial biospheric exchange processes and fossil fuel combustion but are uncoupled for oceanic air-sea gas exchange. High-precision measurements of both species can therefore provide constraints on the carbon cycle and can be used to quantify fossil fuel $CO_2$ (ff$CO_2$) emission estimates. In the case of $O_2$, however, due to its large atmospheric mole fraction of $O_2$ (~20.9 %) it is very challenging to measure small variations to the degree of precision and accuracy required for these applications. We have tested an atmospheric $O_2$ analyser based on the principle of cavity ring-down spectroscopy (Picarro Inc., model G2207-*i*), both in the laboratory and at the Weybourne Atmospheric Observatory (WAO) field station in the UK, in comparisons to well-established, pre-existing atmospheric $O_2$ and $CO_2$ measurement systems.

In laboratory tests analysing air in high-pressure cylinders, from the Allan deviation we calculated a precision of ± 1 ppm (1σ standard deviation of 300 seconds mean), and a 24-hour peak-to-peak range of hourly averaged values of 1.2 ppm. These results are close to atmospheric $O_2$ compatibility goals as set by the UN World Meteorological Organization. From measurements of ambient air conducted at WAO we found that the built-in water correction of the G2207-*i* does not sufficiently correct for the influence of water vapour on the $O_2$ mole fraction. When sample air was pre-dried and employing a 5-hourly baseline correction with a reference gas cylinder, the G2207-*i*'s results showed an average difference from the established $O_2$ analyser of 13.6 ± 7.5 per meg (over two weeks of continuous measurements). Over the same period, based on measurements of a so-called "target tank" (sometimes known as a "surveillance tank"), analysed for 12 minutes every 7 hours, we calculated a repeatability of ± 5.7 ± 5.6 per meg and a compatibility of ± 10.0 ± 6.7 per meg for the G2207-*i*. To further examine the G2207-*i*'s performance in real-world applications we used ambient air measurements of $O_2$ together with concurrent $CO_2$ measurements to calculate ff$CO_2$. Due to the imprecision of the G2207-*i*, the ff$CO_2$ calculated showed large differences from that calculated from the established system, and had a large uncertainty of ± 13.0 ppm, which was roughly double that from the established system (± 5.8 ppm).



## 1 Introduction


Oxygen ($O_2$) is the most abundant molecule in the atmosphere after nitrogen ($N_2$), with an atmospheric background mole fraction of approximately 20.94 % (Tohjima et al., 2005a). Due to this large atmospheric background, $O_2$ measurements are sensitive to changes in the mole fractions of trace gases, such as carbon dioxide ($CO_2$). $O_2$ measurements are therefore typically reported on a relative scale calculated as the change in the ratio of $O_2$ to $N_2$ relative to a standard $O_2/N_2$ ratio, as

given in Eq. (1), and expressed in "per meg" units.

$$\delta \left( \frac{O_2}{N_2} \right) = \left( \frac{O_2/N_{2\ sample} - O_2/N_{2\ reference}}{O_2/N_{2\ reference}} \right) \times 10^6 \quad (1)$$

In practice, atmospheric $N_2$ is far less variable than $O_2$ meaning that changes in the $O_2/N_2$ ratios can be assumed to be representative of $O_2$ mole fraction (Keeling and Shertz, 1992). In comparing changes in $O_2$ to changes in $CO_2$, on a mole for mole basis, a 1 per meg change in $O_2$ is equivalent to a 0.2094 ppm (parts per million) change in $CO_2$ mole fraction (Keeling

et al., 1998).

Over the past three decades atmospheric $O_2$ has been decreasing at a rate of ~15 per meg yr$^{-1}$ primarily owing to fossil fuel combustion (Keeling and Manning, 2014). In contrast the atmospheric $CO_2$ mole fraction has increased from approximately 277 ppm at the beginning of the industrial era to 410 ppm in 2019 (Friedlingstein et al., 2020), also predominantly due to fossil fuel combustion. For most processes that cause variability in atmospheric $O_2$, there is an anti-correlated change in

atmospheric $CO_2$, therefore high-precision measurements of atmospheric $O_2$ play an increasingly important role in our understanding of atmospheric $CO_2$, carbon cycling, and other biogeochemical processes (Pickers et al., 2017, e.g. Resplandy et al., 2019, Battle et al., 2019, Tohjima et al., 2019). Fluxes of $O_2$ and $CO_2$ in and out of the atmosphere are strongly coupled for terrestrial biosphere exchange with a global average oxidative ratio (OR) in the range of 1.03 to 1.10 mol mol$^{-1}$ (Severinghaus, 1995). For fossil fuel combustion, dependent on fuel type, the OR is in the range of 1.17 to 1.95 mol mol$^{-1}$

(Keeling, 1988b). Whereas $O_2$ and $CO_2$ fluxes are uncoupled for oceanic air-sea gas exchange primarily due to inorganic reactions in the water involving the carbonate system and not $O_2$, as well as differences in air-sea equilibration times between the two gases.

The relationship between $O_2$ and $CO_2$ fluxes has also allowed for the derivation of the tracer "atmospheric potential oxygen" (APO), as defined in Eq. (1) (Stephens et al., 1998).

$$APO \approx O_2 + (1.1 \times CO_2) \quad (2)$$

Where the factor 1.1 represents the mean value of the $O_2$:$CO_2$ OR for terrestrial biosphere photosynthesis and respiration (Severinghaus, 1995). APO is therefore, by definition, invariant with respect to the terrestrial biosphere. Changes in APO therefore mainly reflect changes in ocean-atmosphere exchange of $O_2$ and $CO_2$ (primarily on seasonal and longer timescales), with a contribution from fossil fuels on both shorter and longer timescales. APO can thus be used to examine

oceanic $CO_2$ fluxes and to quantify fossil fuel $CO_2$ (ff$CO_2$) emissions (Pickers et al., 2022).

The World Meteorological Organization (WMO) Global Atmospheric Watch (GAW) programme has established a compatibility goal for $O_2$ of ± 2 per meg (± 0.4 ppm), which is the scientifically desirable level of compatibility required to



resolve latitudinal gradients and long-term trends (Crotwell et al., 2019). There is also an extended goal of ± 10 per meg (± 2 ppm) which is suitable for some specific applications when expected variation are relatively large, such as fossil fuel quantification in large cities (Crotwell et al., 2019). In order to be able to meet the WMO compatibility goals, it is recommended that a measurement system's analytical precision should not exceed half of the compatibility (i.e., ± 1 per meg, ± 0.2 ppm), however, routinely achieving a measurement precision of ± 1 per meg, is not yet achievable for the majority of laboratories and field stations making high-precision measurements of atmospheric $O_2$. The large atmospheric background of $O_2$ makes it extremely challenging to measure the relatively small variations to the level of precision required, since measuring a of 0.2 ppm against the background (~209400 ppm) requires a relative precision of 0.0001 %.

Presently, there are several different analytical techniques available for measuring atmospheric $O_2$ to a high precision: interferometry (Keeling, 1988a), isotope ratio mass spectrometry (Bender et al., 1994), paramagnetic techniques (Manning et al., 1999), vacuum ultraviolet absorption (VUV) (Stephens et al., 2011), gas chromatography (Tohjima, 2000), and electrochemical fuel cells (Stephens et al., 2007). The most precise of these current methods is the VUV absorption technique however, VUV $O_2$ analysers are "homemade" and are not commercially available thus limiting their widespread applications. None of these techniques are "off-the-shelf" instruments, all of them are complex and time-consuming systems to design, build, and optimise, with very precise pressure, temperature, and flow control needed. All of the techniques also require frequent interruption to sample measurement to carry out calibration procedures (Kozlova and Manning, 2009). The supply of calibration gases for such systems is particularly labour intensive, both due to their relatively rapid consumption rate and that no commercial gas supply company is able to provide suitable gas mixtures for atmospheric $O_2$ research. Accurate, high-precision atmospheric $O_2$ measurements therefore remain challenging. An alternative commercially available $O_2$ analyser with less requirements for external gas handling, air-sample drying, and calibration procedures could consequently revolutionise the field of atmospheric $O_2$ measurements if the required performance could be achieved and if it were relatively easy to operate with low maintenance requirements and a lower rate of calibration gas consumption.

In this paper we present the results from the analysis of a Picarro Inc. G2207-*i* Oxygen analyser, which operates on the principle of cavity ring-down spectroscopy technology (CRDS) (hereafter referred to as the G2207-*i*) and evaluate its performance in comparison to established $O_2$ measurement systems in the University of East Anglia (UEA) Carbon Related Atmospheric Measurements (CRAM) Laboratory and at the Weybourne Atmospheric Observatory (WAO; North Norfolk, UK). Unlike most other analytical techniques used for atmospheric $O_2$ measurements, the G2207-*i* does not require a continuous reference gas supply, has built-in pressure and flow control, and has the potential for reduced sample drying requirements due to a built-in water measurement and correction procedure. These features make the G2207-*i* a potentially desirable analyser for high-precision atmospheric $O_2$ research, but we note that it would still require the same rigorous calibration procedures as other analysers (Kozlova and Manning, 2009), albeit possibly at reduced frequency. The accuracy, precision, and drift are quantified and presented here in the context of WMO/GAW guidelines (Crotwell et al., 2019). In order to further examine the performance of the G2207-*i* in real-world applications, we also calculated ff$CO_2$ from concurrent $O_2$ and $CO_2$ measurements made, using the novel methodology presented by Pickers et al. (2022). We compare



ffCO$_2$ calculated with O$_2$ measurements from the G2207-$i$ installed at WAO with ffCO$_2$ calculated from the established O$_2$ system employing a Sable Systems International Inc. "Oxzilla II" fuel cell analyser.

## 2 Methods

### 2.1 Picarro G2207-$i$ O$_2$ analyser

The Picarro G2207-$i$ O$_2$ analyser measures the mole fractions of the two most abundant atmospheric O$_2$ isotopologues, $^{16}O^{16}O$ and $^{16}O^{18}O$, through absorption spectra at 7882.18670 cm$^{-1}$ and 7882.050155 cm$^{-1}$, respectively (Berhanu et al., 2019). The design principles of this analyser have been described in detail by Berhanu et al. (2019). In our study we evaluate only what is called the "O$_2$ concentration" mode, measuring only the $^{16}O^{16}O$ isotopologue. In the other mode, called the "$\delta^{18}O$ plus O$_2$ concentration" mode, O$_2$ mole fraction values are considerably less precise, as the analyser is not optimised for $^{16}O^{16}O$ measurements (primarily via a different set point for the pressure in the cavity). The analyser reports both "wet" and "dry" O$_2$ mole fraction values. The "wet" values (O$_{2,NC}$) do not have any correction applied to them, whereas the "dry" values (O$_{2,WC}$) are corrected for the dilution effect of water vapour on the O$_2$ mole fraction, as well as spectroscopic interference, using the analyser's parallel water vapour mole fraction measurements.

### 2.2 CRAM laboratory measurement of cylinder gases

The performance of the G2207-$i$ was evaluated in the UEA CRAM Laboratory by measuring a suite of 12 gas cylinders all containing dry natural air with varying O$_2$ mole fractions. The cylinders were stored horizontally in a thermally insulated "Blue Box" enclosure in order to prevent gravitational and thermal fractionation of O$_2$ relative to N$_2$ (Keeling et al., 2007). The O$_2$ composition of each of these cylinders was precisely defined on the Scripps Institution of Oceanography (SIO) O$_2$ scale (Keeling et al., 2007) using a VUV O$_2$ analyser, also in the CRAM Laboratory. The CO$_2$ mole fraction was defined on the "WMO CO$_2$ X2007" scale (Zhao and Tans, 2006) using a Siemens Corp. Ultramat model 6F non-dispersive infrared (NDIR) CO$_2$ analyser. Five of these cylinders were working secondary standards (WSSes) which were used to calibrate the G2207-$i$, one was a reference tank (RT; explained below in section 2.3.2), and the other six were treated as cylinders with unknown mole fractions (Table 1). The six "unknown" cylinders were used to evaluate the performance of the analyser with a CO$_2$ mole fraction range of 375 to 443 ppm and an O$_2$/N$_2$ ratio range of -915 to 435 per meg, a much larger range than would typically be observed in ambient air.

The cylinders were run consecutively, starting with the six "unknowns" and ending with the 5 WSSes, with the RT run at the beginning and end, this was repeated twice. Each of the gas cylinders was flushed for 20 minutes prior to running on the G2207-$i$ to allow for removal of stagnant air and equilibration of the pressure regulators; air from each cylinder was then passed through the analyser for 20 minutes, with the first 8 minutes of data discarded to allow flushing of the previous cylinder's air from the cavity. The remaining 12 minutes for each cylinder was then averaged to give the "raw" O$_{2,NC}$ value for each cylinder as measured on the G2207-$i$.

**Table 1. Declared $O_2/N_2$ ratios and $CO_2$ mole fractions with ± 1σ standard deviations of the five WSSes, RT, and six "unknown" cylinder gases used in the CRAM Laboratory tests of the G2207-$i$**

| Cylinder number | Cylinder ID | Declared $O_2$ (per meg)[a] | Declared $CO_2$ (ppm)[b] |
|---|---|---|---|
| WSS1 | D089507 | -565.5 ± 1.3 | 428.741 ± 0.018 |
| WSS2 | D801299 | -486.1 ± 3.0 | 381.230 ± 0.016 |
| WSS3 | D073409 | -658.4 ± 2.2 | 398.875 ± 0.018 |
| WSS4 | D073419 | -926.4 ± 5.9 | 440.355 ± 0.072 |
| WSS5 | D073418 | -782.7 ± 5.6 | 413.662 ± 0.057 |
| RT | CC78691 | -414.3 ± 0.8 | 384.915 ± 0.005 |
| 1 | D273555 | -914.8 ± 0.7 | 443.384 ± 0.013 |
| 2 | D399093 | -880.5 ± 0.9 | 415.246 ± 0.003 |
| 3 | ND29112 | -582.0 ± 1.0 | 399.976 ± 0.004 |
| 4 | ND29110 | -375.0 ± 1.3 | 381.544 ± 0.004 |
| 5 | D273559[c] | 411.7 ± 2.1 | 375.122 ± 0.007 |
| 6 | D801298[c] | 434.6 ± 0.3 | 412.934 ± 0.002 |

[a] Values declared with a VUV $O_2$ analyser in the CRAM Laboratory traceable to the SIO $O_2$ scale

[b] Values declared with a Siemens Ultramat 6F NDIR $CO_2$ analyser in the CRAM Laboratory traceable to the WMO $CO_2$ X2007 scale

[c] The $O_2$ values of these cylinders is far outside the range observed in ambient air, thus are less relevant to the applications of atmospheric observations but have been included in this analysis for completeness of examining the analysers performance.

The G2207-$i$ has a linear response to $O_2$ mole fraction (Eq. (3))

$$y = Bx + C \qquad (3)$$

where, B and C are the coefficients derived from the slope and intercept of the linear regression calculated from the measurement of the WSSes. Therefore, a minimum of two WSS cylinders are required to determine the B and C coefficients, but by using five we are able to calculate the coefficient of determination ($R^2$), as well as providing more robustness in the fit. The calibration equation was used to convert the "raw" $O_{2,NC}$ values taken from the G2207-$i$ ("x" in Eq.(3)) into what we call "ppm equivalent" (ppmEquiv) $O_2$ units ("y" in Eq. (3)), as described in Kozlova and Manning (2009). A linear interpolation between the RT at the beginning and end of each run was used as a baseline for the run and subtracted from all other cylinder measurements to correct for short-term instrumental variations. The calibration curve (Eq. (3)) for the G2207-$i$ was also determined relative to the interpolated RT values (WSS - RT), thus all the unknown cylinder measurements could be converted into ppmEquiv. The ppmEquiv $O_2$ units were then converted to per meg units, providing a δ($O_2/N_2$) value for each "unknown cylinder, using Eq. (4).



$$\delta\left(\frac{O_2}{N_2}\right) = \frac{\delta O_2 + (CO_2 - 363.29) \times S_{O2}}{S_{O2} \times (1 - S_{O2})} \quad (4)$$

where, $\delta O_2$ is the calibrated G2207-$i$ $O_{2,NC}$ values in ppmEquiv units, $CO_2$ is the declared cylinder $CO_2$ mole fraction from
the Siemens analyser in ppm, $S_{O2}$ is 0.2094 which is the standard mole fraction of $O_2$ molecules in dry air, and 363.29 is an
arbitrary $CO_2$ reference value in ppm, inherent to the SIO $O_2$ scale (Stephens et al., 2007).

### 2.3 Weybourne Atmospheric Observatory field tests

Weybourne Atmospheric Observatory (WAO) is located on the north Norfolk coast, UK ((52°57'02''N, 1°07'19''E),
approximately 35 km north-northwest of Norwich, 170 km northeast of London and 200 km east of Birmingham. It is part of
the European Union's Integrated Carbon Observation System (ICOS) and the World Meteorological Organization's (WMO)
Global Atmosphere Watch (GAW) programme. High-precision, high-accuracy, continuous measurements of a wide array of
atmospheric gas species (including greenhouse gases, isotopes, reactive gases) are carried out at a fine temporal scale,
funded in part through the UK's National Centre for Atmospheric Science (NCAS) long-term measurement programme.

Atmospheric $O_2$ and $CO_2$ have been measured continuously at WAO since 2008 (Wilson, 2013). $O_2$ is measured with a
"Oxzilla II" $O_2$ analyser (Sable Systems International Inc.) (hereafter referred to as the "Oxzilla"), and $CO_2$ is measured with
an Ultramat 6E NDIR analyser (Siemens Corp.). These analysers are in series, with the air sample first passing through the
Ultramat 6E and then the Oxzilla, with rigorous gas handling and calibration protocols followed (as in Stephens et al., 2007).
The G2207-$i$ was installed at WAO from 23 October 2019 – 02 November 2019, sampling from a solar shield aspirated air
inlet (AAI) at a height of 10 m above ground level (AGL) (20 m above sea level (ASL)). The AAI protects the inlet from
solar radiation and generates a continuous air flow over the inlet, thus preventing the differential fractionation of $O_2$
molecules relative to $N_2$ molecules due to ambient temperature variations (Blaine et al., 2006) and relatively slow inlet flow
rates (Manning, 2001). A full plumbing diagram of the gas-handling set-up at WAO is displayed in Fig. 1.





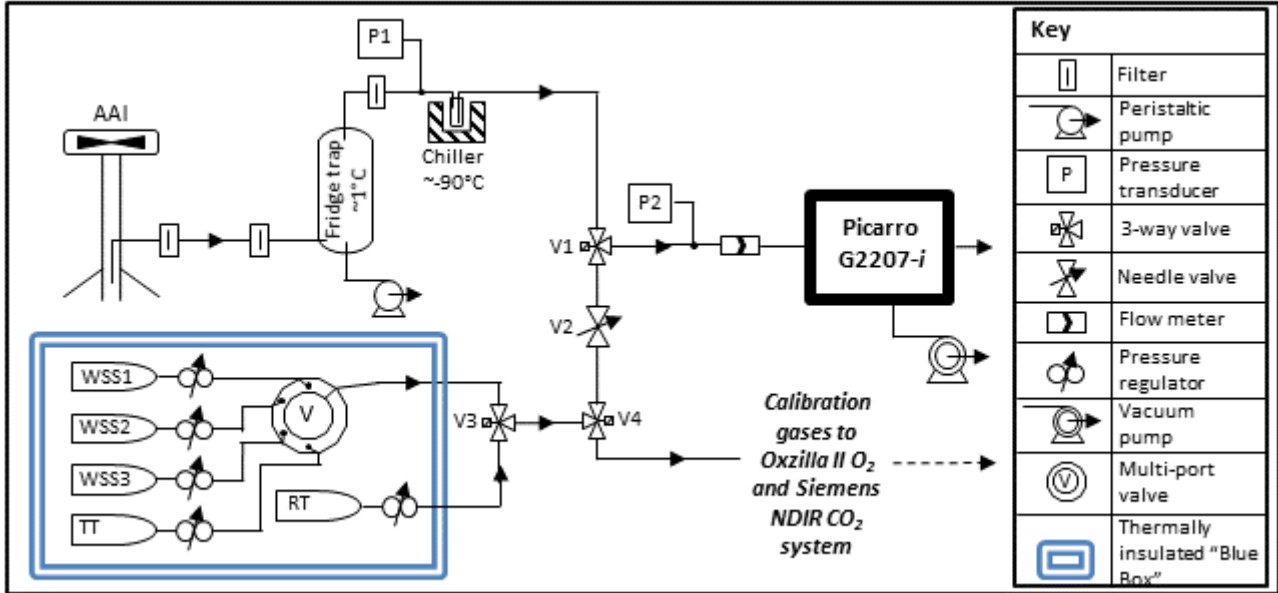

**Figure 1: Gas handling plumbing diagram of the Picarro G2207-*i* installed at WAO. (AAI, aspirated air inlet; WSS, working**
**secondary standard; RT, reference tank; TT, target tank). Calibration gases were shared with the established $O_2$ and $CO_2$ system**
**(using V4), but air was sampled through separate AAIs.**

### 2.3.1 Drying

Water vapour mole fractions in the troposphere vary from a few ppm to a few percent over small temporal and spatial scales,
this water vapour has a diluting effect on atmospheric gas measurement. A 1 ppm increase of water vapour will dilute the
measured atmospheric $O_2$ by approximately 1.3 per meg (Stephens et al., 2007); the existing method for high-precision
atmospheric $O_2$ measurements is therefore to dry the sample air to less than 1 ppm before measurement in order to prevent
the dilution effect of water vapour. All calibration and RT gases are also dried to less than 1 ppm water vapour. Furthermore,
measurements using spectroscopy techniques are also sensitive to water vapour variability due to changes in the degree of
pressure broadening of the spectroscopic lines used to measure the $O_2$ and $\delta^{18}O_2$. Water vapour correction has previously
been successfully implemented for measurements of $CO_2$ and methane ($CH_4$) with CRDS analysers (Chen et al., 2010);
however, in order to achieve accuracies within the WMO goal of 1% $H_2O$ custom coefficients must be obtained for each
analyser (Rella et al., 2013).

As discussed in section 2.1, $O_2$ measurements are reported by the G2207-*i* as "wet" ($O_{2,NC}$) and, after the implementation of
water correction, "dry" ($O_{2,WC}$). In order to evaluate the effectiveness of the built-in water correction procedure for
compensating for water vapour dilution, ambient air was sampled with three different drying regimes: no drying, partial
drying, and full drying. Under the full drying conditions (which is the current standard practice), the sample air passed
through a fridge trap (~1°C) and a cryogenic chiller trap (~-90°C), removing water vapour to < 1 ppm. Under partial drying
the chiller was bypassed, so the sample air only passed through the fridge trap which dries the air to approximately 5000





ppm of water vapour. With no drying, both the chiller and fridge were bypassed. Air was simultaneously sampled through a
separate AAI (10 m AGL) into the pre-existing $O_2$ and $CO_2$ system with full drying during each of these stages. The time
difference between air travelling from the AAIs to each of the two analysers was accounted for.

To evaluate the built-in water correction procedure of the G2207-*i* the $O_{2,WC}$ values were compared with measurements from
the Oxzilla (which was continuously sampling fully dried air) for the no drying and partial drying periods, and the $O_{2,NC}$ and
$O_{2,WC}$ G2207-*i* values were compared to the Oxzilla when sampling fully dried air.

### 2.3.2 Calibration procedure

A tailor-made calibration protocol was developed for the G2207-*i* following ICOS atmospheric station specifications (ICOS-
RI, 2020). The calibration cylinders were stored horizontally in a thermally insulated "Blue Box" enclosure in order to
prevent gravitational and thermal fractionation of $O_2$ and $N_2$. The calibration gases consisted of three WSSes with precisely
defined $O_2$ and $CO_2$ values which span the unpolluted atmospheric range (traceable to the SIO $O_2$ and WMO $CO_2$ X2007
scales) and a reference tank (RT) with $O_2$ and $CO_2$ values close to ambient air conditions at the site. The repeatability and
compatibility of the analyser were evaluated using a target tank (TT) with precisely defined $O_2$ and $CO_2$ values. With full
drying of the sample air each of the WSSes, the RT, and the TT were run for 20 minutes, the first 8 minutes was discarded
due to the sweep-out time of the G2207-*i*, and the final 12 minutes averaged to determine the cylinder value for the given
run. Under partial and no drying the run-time of the cylinders was increased in order to fully flush the G2207-*i* of water
vapour; each cylinder was therefore run for 32 minutes, with the first 20 minutes being discarded and the final 12 minutes
averaged.

A full 3-gas WSS calibration of the G2207-*i* was run every 23 hours, this frequency is intentionally not a multiple of 24
hours in order to prevent aliasing the data by calibrating under environmental conditions that may occur at the same time
each day. This calibration corrects for drift in the span or non-linearity of the analyser. As in the CRAM laboratory tests (see
section 2.2), the WSSes were used to define a calibration equation to convert the raw analyser $O_2$ values into ppmEquiv $O_2$
units. Eq. (3) and the concurrent $CO_2$ measurement from the Ultramat 6E NDIR were then used to convert this into per meg
units.

The RT is used for data correction caused by short-term instrument drift and was run every 5 hours. A linear interpolation
between each of the RT run averages was treated as a baseline and subtracted from all subsequent air and cylinder
measurements. The calibration curve for the G2207-*i* was also determined relative to the RT values (WSS- RT) , thus the air
measurement differences can be easily converted into per meg units.

Finally, the TT was run every 7 hours, this cylinder is used to quantify the repeatability and compatibility of the analyser.
"Repeatability" is defined as the closeness of agreement between results of successive measurements of the same measure
carried out under the same measurement conditions and is considered as a proxy for the precision of a measurement system.
"Compatibility" is defined as the averaged $O_2$ value of all TT runs over time, compared to the values declared by the VUV,
and provides a measure of the compatibility to the SIO scale over time (Kozlova and Manning, 2009). The TT air does not



pass through the AAI or drying lines (Fig. 1) so it is therefore mainly representative of the analyser's repeatability and compatibility only.

## 2.5 Quantifying fossil fuel CO₂ using atmospheric potential oxygen

In order to further assess the G2207-*i*'s performance in real-world applications the $O_{2,NC}$ observations from the full drying regime period at WAO were used to isolate the fossil fuel component of the concurrent $CO_2$ observations and then compared to the $ffCO_2$ values calculated from atmospheric potential oxygen (APO) derived from the Oxzilla $O_2$ observations following the methodology outlined in Pickers et al. (2022).

The tracer APO, derived by Stephens et al. (1998), was first calculated using Eq. (4) (using both G2207-*i* $O_{2,NC}$ and Oxzilla
$O_2$ values); these APO values were then used to calculate $ffCO_2$ using Eq. (5).

$$APO = [O_2] + \left(\left(\frac{-1.1}{0.2094}\right) \times (350 - [CO_2])\right) \quad (4)$$

where $O_2$ and $CO_2$ are in per meg and ppm units, respectively; -1.1 is the global average $O_2$:$CO_2$ terrestrial biosphere-atmosphere exchange rate (Severinghaus, 1995), 0.2094 is the mole fraction of $O_2$ molecules in dry air (Tohjima et al., 2005b), and 350 is an arbitrary reference value for $CO_2$ in ppm. Multiplying $CO_2$ by -1.1 and dividing by 0.2094 converts the
$CO_2$ data from ppm to per meg units.

$$ffCO_2 = \frac{APO - APO_{bg}}{R_{APO:CO2}} \quad (5)$$

Where APO is derived from Eq. (4) in per meg units, $APO_{bg}$ is the APO background, or baseline, value determined using a statistical baseline fitting procedure, and $R_{APO:CO2}$ is the APO:$CO_2$ combustion ratio for fossil fuel emissions. The $APO_{bg}$ values were determined using the rfbaseline function from the IDPmisc package in R, which implements robust fitting of
local regression models, with a smoothing window of one week (Ruckstuhl et al., 2012). The APO:$CO_2$ emission ratio ($R_{APO}$) used is -0.3 mol mol⁻¹, an approximate mean value for WAO as determined from the COFFEE inventory (given that the APO:$CO_2$ ratio = $O_2$:$CO_2$ + 1.1) (Pickers, 2016, Steinbach et al., 2011). The uncertainty on the $ffCO_2$ mole fractions was calculated using Eq. (5) with the upper and lower uncertainty limit for each variable (where the measurement uncertainty for APO was calculated by summing in quadrature the $CO_2$ and $O_2$ measurement uncertainty for each analyser), then taking the
SD of the resultant $ffCO_2$ value for each combination for each hourly time stamp.

## 3. Results and discussion

### 3.1 Precision and drift

To assess the short-term precision and optimal averaging time of the G2207-*i* the Allan deviation technique (Werle et al., 1993) was used whilst sampling a compressed-air cylinder in the laboratory (50 L, 200 bar). The cylinder was run for 24
hours with a sample flow rate of 94 mL/min and cavity pressure and temperature of 255 torr and 45°C. The results of this





Allan deviation analysis are in agreement with those obtained by Berhanu et al. (2019), where a precision of 1 ppm (~4.8 per meg) was achieved after an averaging time of 300 seconds, and continues to improve until around 3000 seconds where a precision of ~0.3 ppm (~1.44 per meg) is reached (Figure 2).

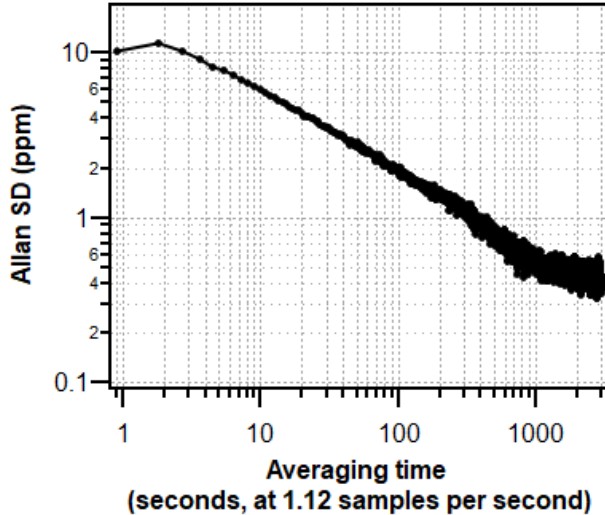

**Figure 2. Allan deviation plot displaying the precision of the G2207-*i* O$_2$ mole fraction measured from an ambient compressed-air cylinder.**

To evaluate the analyser drift (i.e., the changing sensitivity of the analysers response with time), O$_{2,NC}$ values from the G2207-*i* were averaged to 1 hour (Fig. 3b; reported in ppm where 1 ppm corresponds to a change of 4.8 per meg in the O$_2$/N$_2$ ratio). The G2207-*i* datasheet states a maximum drift at STP (over 24 hours, peak-to-peak, 1-hour internal average at 21 %

O$_2$) of <6 ppm. We found that over 24 hours, the maximum peak-to-peak drift of the hourly averages is ~1.2 ppm (approximately 5.76 per meg); this is better than stated by Picarro Inc. but does not meet the WMO compatibility goal of ± 2 per meg, as the internal drift of the analyser is greater than this goal. The standard deviation of each of these hourly averages is ~14.5 ppm (~69.6 per meg) (Fig. 3a), this is caused by the large amount of analyser noise in the raw 1 second data points, spanning ~100 ppm (~480 per meg) (Fig. 3c). The overall drift over the 24 hours of raw data however is very small, shown

by a linear regression slope of -4.26 x 10$^{-6}$ (Fig. 3c).



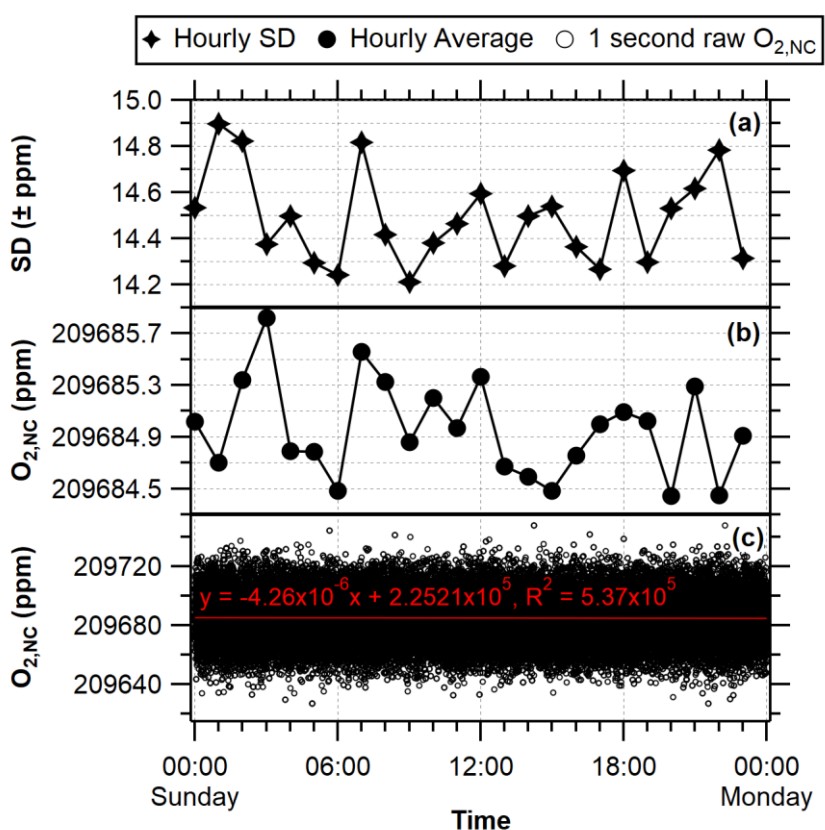

**Figure 3.** $O_{2,NC}$ **mole fractions from the G2207-*i* sampling dry compressed cylinder air over 24 hours, reported in ppm, where 1 ppm corresponds to a change of 4.8 per meg in the $O_2/N_2$ ratio. (a) Standard deviation of the hourly averaged values. (b) Hourly averaged $O_{2,NC}$. (c) Raw 1 second $O_{2,NC}$ values, the red line depicts the linear regression line, with the equation and $R^2$ value written above.**

### 3.2 CRAM laboratory measurement of cylinder gases

The G2207-i analyser performance was evaluated by measuring six gas cylinders with precisely defined O2 and CO2 values as measured on a VUV O2 analyser and Siemens Ultramat 6F NDIR CO2 analyser (Table 1). The difference between the O2,NC values (per meg) as measured by the G2207-i and the declared values from the VUV are shown in Table 2, for both runs with and without the RT interpolation applied.

For both runs without the application of the RT interpolation the difference between the VUV declared value and that measured by the G2207-i is very large, and far outside of an acceptable range (Table 2), with an average difference from the declared values for all cylinders of 22.0 ± 10.3 per meg. For all cylinders, except for cylinder 5 and 6, a large improvement in the difference is seen after the application of the RT correction. Due to the large differences between the declared and measured values without the RT correction applied, only the results with the RT correction will be discussed hereafter.



**Table 2. The difference between the O₂ value of each cylinder as measured on the G2207-*i* and the VUV analyser (G2207-*i* - VUV), for two runs on the G2207-*i* with and without RT correction applied.**

| Cylinder no. | Declared O2 (per meg) | Without RT correction | | | With RT correction | | |
|---|---|---|---|---|---|---|---|
| | | Run 1 difference from declared (per meg)[a] | Run 2 difference from declared (per meg)[a] | Mean of absolute differences of both runs (per meg)[b] | Run 1 difference from declared (per meg)[a] | Run 2 difference from declared (per meg)[a] | Mean of absolute differences of both runs (per meg)[b] |
| 1 | -914.8 ± 0.7 | 9.9 ± 8.4 | 21.4 ± 8.2 | 15.7 ± 8.1 | 0.4 ± 8.5 | 2.4 ± 8.1 | 1.4 ± 1.4 |
| 2 | -880.5 ± 0.9 | 13.7 ± 8.7 | 26.5 ± 8.3 | 20.1 ± 9.1 | 6.1 ± 8.4 | 7.6 ± 8.2 | 6.9 ± 1.1 |
| 3 | -582.0 ± 1.0 | 8.1 ± 8.5 | 22.4 ± 11.3 | 15.3 ± 10.1 | 0.7 ± 8.0 | 3.1 ± 11.2 | 1.9 ± 1.7 |
| 4 | -375.0 ± 1.3 | 12.4 ± 11.6 | 18.4 ± 9.5 | 15.4 ± 4.2 | 5.8 ± 11.3 | -1.1 ± 9.5 | 3.5 ± 3.3 |
| 5 | 411.7 ± 2.1 | 44.0 ± 12.6 | -3.6 ± 11.5 | 23.8 ± 28.6 | 19.0 ± 12.4 | -40.1 ± 10.2 | 29.6 ± 14.9 |
| 6 | 434.6 ± 0.3 | 44.6 ± 5.4 | -39.1 ± 10.2 | 41.9 ± 3.9 | 22.2 ± 5.1 | -49.8 ± 11.5 | 36.0 ± 19.5 |

[a] ± 1σ standard deviation of the 12-minute G2207-*i* average.

[b] ± 1σ standard deviation of the average of the run 1 and run 2 G2207-*i* - VUV difference.


Cylinders 5 and 6 contain an O₂ values far higher than that found in ambient air (411.7 and 432.6 and per meg, respectively) and outside of the range spanned by the WSSes used for calibration. For these two cylinders, the difference between the declared value and that measured by the G2207-*i* is far larger than the other cylinders and also more variable between the two runs with a standard deviation of the absolute values between the two runs of 14.9 and 36.0 per meg, respectively (Table 2). Berhanu et al. (2019) found that the accuracy of the G2207-*i* was reduced when the $CO_2$ mixing ratio was much higher than that of ambient air but did not observe the same reduction in accuracy with high O₂ mixing ratios. Ignoring the two cylinders with positive O₂, the average absolute difference between the remaining 4 unknown cylinders and the declared values over the two runs is 3.4 ± 2.5 per meg, this is slightly greater than the WMO compatibility goal of ± 2 per meg but does fall within the extended goal of ± 10 per meg and is similar to what can be achieved with an Oxzilla II (Pickers et al., 2017). There is also no correlation between the accuracy and the declared O₂ value excluding the two cylinders with positive O₂ ($R^2 = 0.07$ for run 1, $R^2 = 0.53$ for run 2).

Although the accuracy of the O₂ values measured by the G2207-*i* for these cylinders is variable, particularly for the cylinders with high O₂, the standard deviation of the 2-minute data points used to calculate the final cylinder O₂ value as defined by the G2207-*i* within each run is more consistent. However, the repeatability, used as a proxy for precision, and defined here as the ± 1σ standard deviation of the average of the two measurements of each cylinder are variable. For the two cylinders with high O₂ (cylinders 5 and 6) the repeatability is more than 5 times greater than the WMO extended repeatability goal of ± 5




per meg. For the remaining four cylinder the repeatability is far lower, with cylinder 1 and cylinder 3 both falling within the extended repeatability goal.

### 3.3 Weybourne Atmospheric Observatory field tests

**3.3.1 Partial and no drying of ambient air measurements**

The results from no drying and partial drying of the sample air into the G2207-*i* at WAO are displayed in Fig. 4 and Fig. 5, respectively. The $O_2$ mole fractions reported in ppm units by the G2207-*i* were converted to per meg units using the calibration equations produced through the measurement of the three WSS cylinders every 23 hours, and the concurrent $CO_2$ observations.

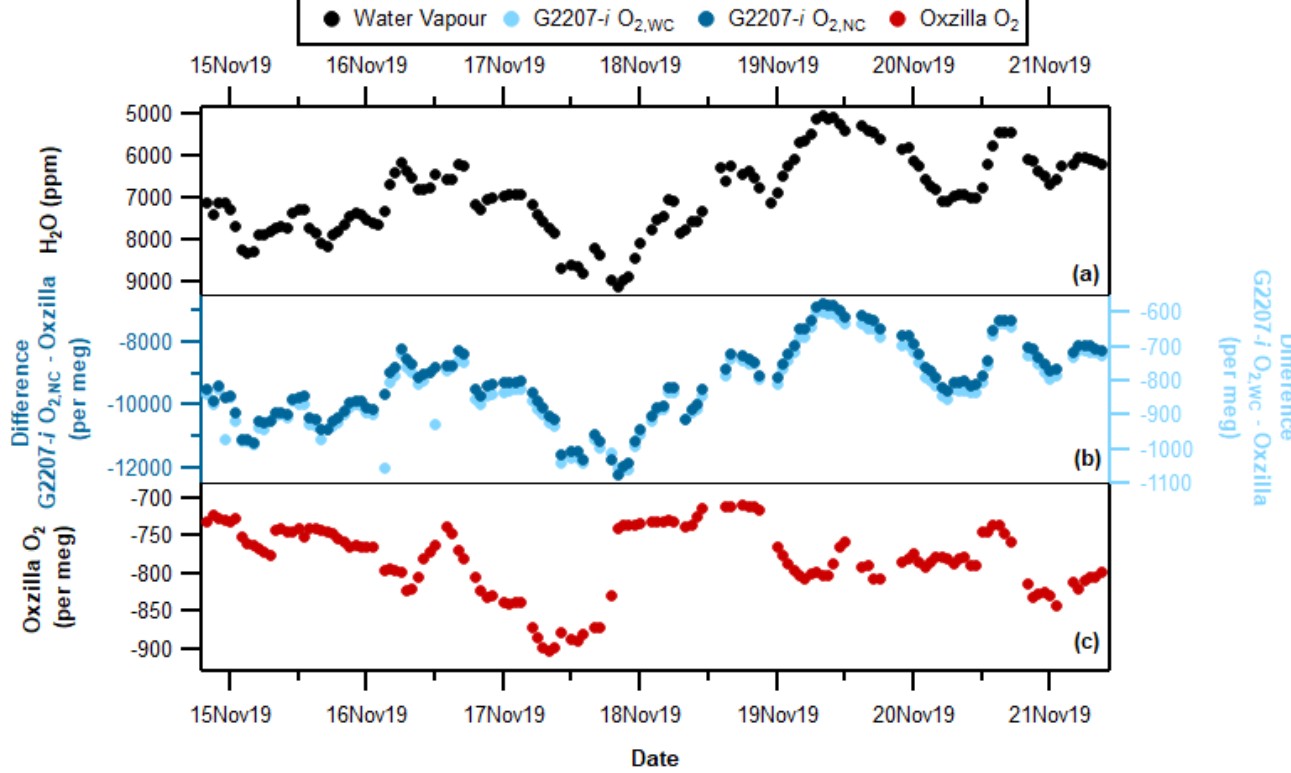


**Figure 4. (a) Hourly averaged water vapour, (b) G2207-*i* – Oxzilla difference for $O_{2,NC}$ (dark blue) and $O_{2,WC}$ (light blue), and (c) Oxzilla $O_2$ with no drying of the sample air through the G2207-*i*. N.B., the reversed water vapour axis and different axis scales for $O_{2,NC}$ and $O_{2,WC}$.**

During the period where there was no drying of the G2207-*i* air sample there is a significant difference between the $O_2$

values reported by the Oxzilla (dried air) and the G2207-*i* $O_{2,NC}$ values (Fig. 4b), this is to be expected due to the diluting effect of water vapour; however, there is also a significant difference between the Oxzilla $O_2$ and the G2207-*i* $O_{2,WC}$ values. Over the entire no drying period the average difference between the Oxzilla observations and the G2207-*i* $O_{2,NC}$ is -9654.41





± 272.84 per meg. The average difference between the Oxzilla and the G2207-*i* $O_{2,WC}$ values is -849.78 ± 31.12 per meg.
Although the difference is substantially smaller with the application of the G2207-*i* built-in water correction procedure, it is
still unusably large, with no similarity in the Oxzilla and G2207-*i* signals and both the $O_{2,NC}$ and $O_{2,WC}$ G2207-*i* values
correlating with the $H_2O$ variability (Fig. 6c and d). This demonstrates that the algorithm currently applied for water
correction is unsuitable for precise $O_2$ measurement.

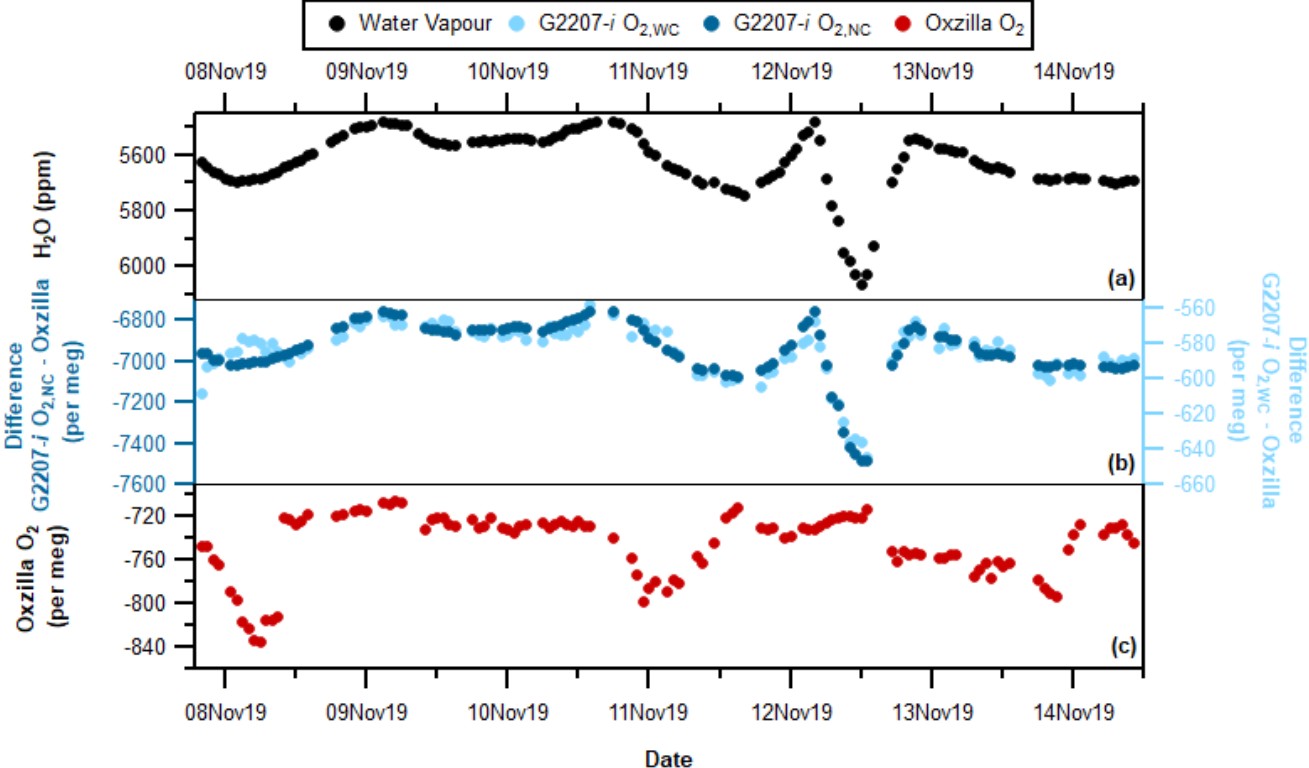

**Figure 5. (a) Hourly averaged Water vapour (top), (b) G2207-*i* – Oxzilla difference for $O_{2,NC}$ (dark blue) and $O_{2,WC}$ (light blue),**
**and (c) Oxzilla $O_2$ (bottom) with partial drying of the sample air through the G2207-*i* (Oxzilla sample air is fully dried). N.B., the**
**reversed water vapour axis, and different axis scales for $O_{2,NC}$ and $O_{2,WC}$. The spike in water vapour on 12 November 2019 is due**
**to a temporary increase in the temperature of the fridge.**

As seen during no drying of the sample air, there is also a significant difference between the reported $O_2$ values of the
Oxzilla and G2207-*i* under the partial drying regime, for both $O_{2,NC}$ and $O_{2,WC}$ (Fig. 5b). With partial drying the time-series
of the difference between the $O_2$ values of the two analysers is a lot smoother than with no drying, this is due to the fridge
trap removing some of the natural variability in the water vapour mole fraction. Over the entire partial drying period the
average difference between the Oxzilla observations and the G2207-*i* $O_{2,NC}$ is -7144.06 ± 258.60 per meg. The average
difference between the Oxzilla and the G2207-*i* $O_{2,WC}$ values is -612.71 ± 31.77 per meg. There is a large improvement with
the application of the water correction procedure; however, as with the no drying results, the difference in $O_2$ values between





the Oxzilla and G2207-*i* $O_{2,WC}$ are too large to be usable for any application, with the $O_{2,NC}$ and $O_{2,WC}$ values correlating with

the $H_2O$ variability (Fig. 6a and b), and therefore will not be investigated further.

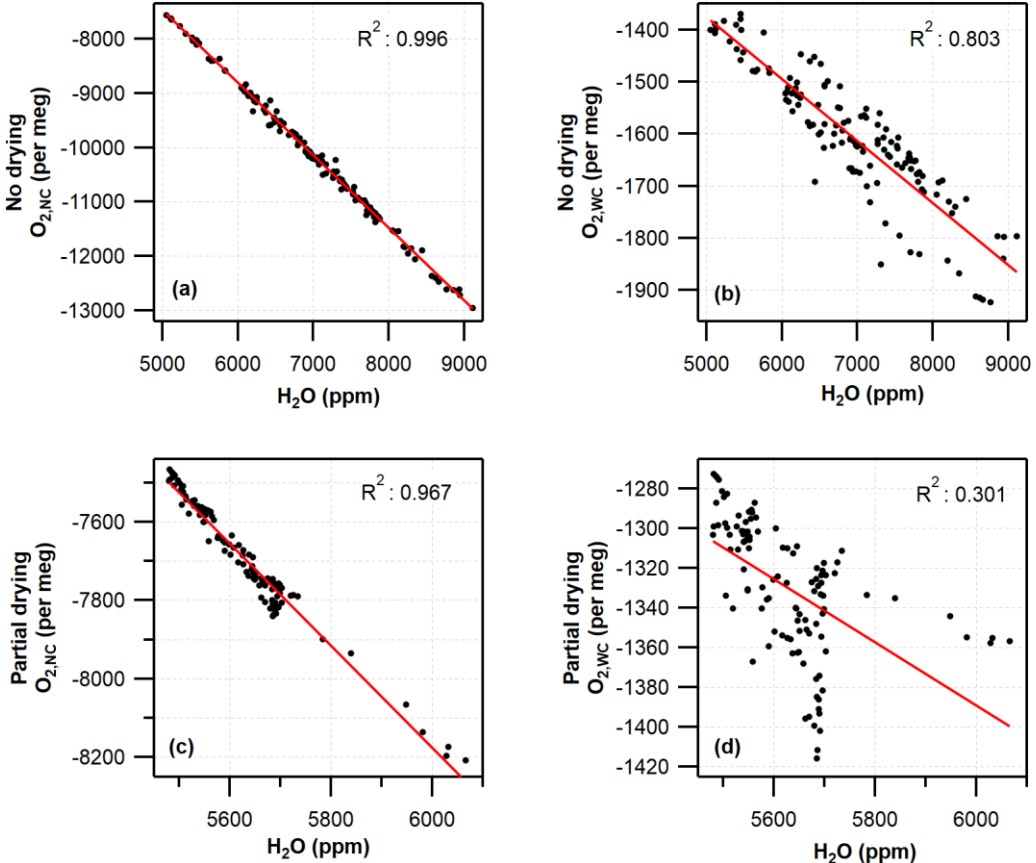

**Figure 6. Correlation between water vapour mole fraction and hourly averaged G2207-*i* $O_2$ for (a) no drying $O_{2,NC}$, (b) no drying $O_{2,WC}$, (c) partial drying $O_{2,NC}$, and (d) partial drying $O_{2,WC}$. Red lines show linear regression.**

Under both "partial drying" and "no drying" regimes, the difference between the Oxzilla and G2207-*i* values is strongly

correlated with the water vapour mole fraction but decreases with the application of the built-in water correction procedure

(Fig. 6). The $R^2$ value decreases from 0.996 to 0.803 for no drying and from 0.967 to 0.301 for partial drying once the water

correction has been applied. Given the correlation between the water vapour mole fraction and the $O_{2,WC}$ reported by the

G2207-*i* these values are not usable without significant improvements to the water correction procedure by Picarro Inc..

Due to the large differences observed between the Oxzilla and G2207-*i* reported $O_2$ values under no drying and partial

drying, no further investigation was undertaken, thus only the fully dried sample air is considered hereafter.





### 3.3.2 Full drying of ambient air measurements

The results from fully drying the sample air between 24 October 2019 and 07 November 2019 are displayed in Fig. 7. The $O_2$ mole fractions reported in ppm units by the G2207-*i* were converted to per meg units using the calibration equations

produced through the measurement of the three WSS cylinders every 23 hours, and the concurrent $CO_2$ observations.

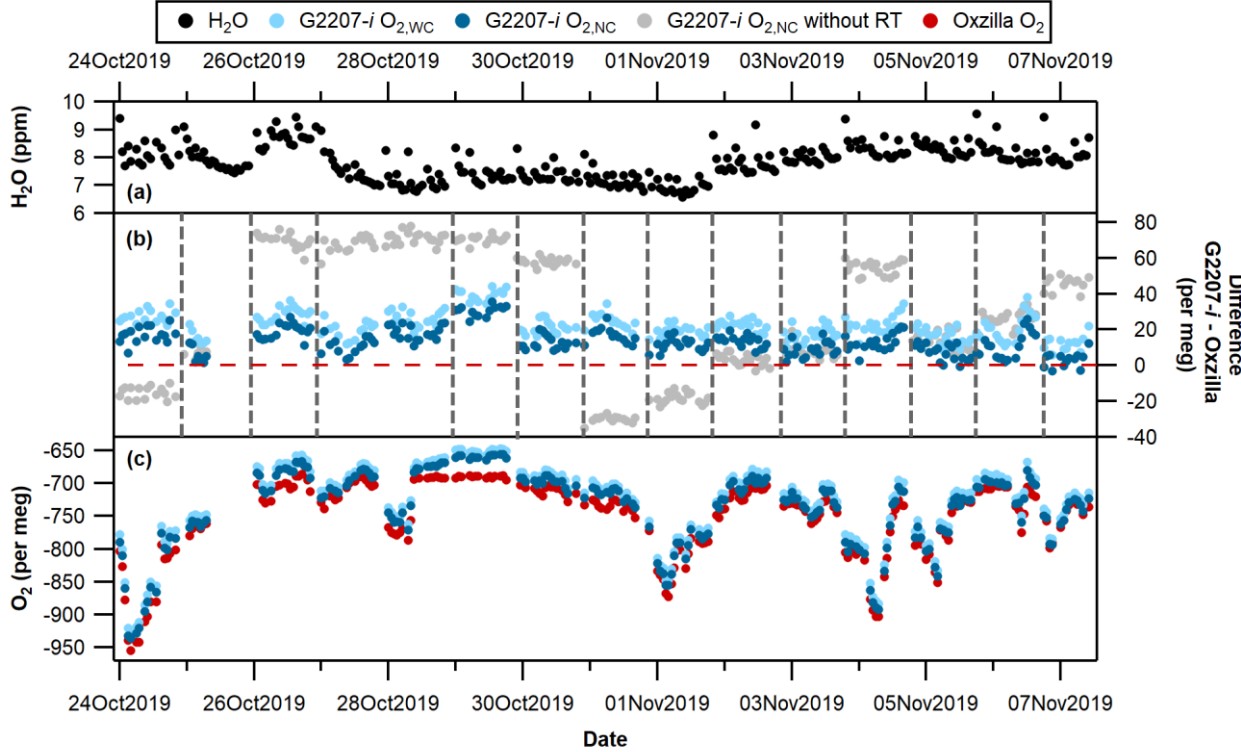

**Figure 7.** Time-series with full drying of the air sample. (a) Hourly averaged water vapour, spikes are due to equilibration after valve switching from cylinder air to sample air. (b) G2207-*i* – Oxzilla difference for $O_{2,WC}$ (light blue), $O_{2,NC}$ (dark blue), and $O_{2,NC}$ without the RT interpolation applied (grey); vertical dashed lines indicated a full 3-gas WSS calibration on the G2207-*i*, and the

red horizontal line indicates zero difference from the Oxzilla. (c) Hourly averaged Oxzilla $O_2$ (red), $O_{2,WC}$ (light blue) and $O_{2,NC}$ (dark blue). Note, there was no WSS calibration on 27 October 2019 due to a macro error which prevented valve switching to calibration gases, therefore the calibration from 26 October 2019 was applied for 46 hours.

There is a greater difference between the Oxzilla and G2207-*i* $O_{2,WC}$ values than the $O_{2,NC}$ values, with an average difference over the entire full drying period of $22.60 \pm 7.41$ per meg compared to $13.59 \pm 7.46$ per meg, respectively. This may be due

to overcorrection of the $O_{2,NC}$ values as the water vapour mole fraction is below the G2207-*i*'s lower detection limit i.e. the G2207-*i* is reporting $H_2O$ mole fractions of approximately 7 ppm (Fig. 7a) (with frequent spikes due to equilibration after switching of V1 (Fig. 1) from cylinder to sample air); however, when the air sample is fully dried by passing through the chiller and fridge trap, the water vapour is reduced to below 1 ppm. This overestimated water correction whilst sampling fully dried air was also found by Berhanu et al. (2019). We therefore only refer to the $O_{2,NC}$ values, which we believe to be

more accurate, in the analysis from now onwards.





The large jumps in the G2207-*i* $O_{2,NC}$ values following WSS calibrations (see Fig. 7b, grey points) are caused by a drift in the instruments baseline; these jumps were reduced through the application of the 5-hour RT interpolation procedure (refer to Section 2.3.2). After the application of the RT interpolation the jumps between WSS calibrations were vastly reduced (see Fig. 7), thus the ffCO$_2$ results in section 3.5 have this applied.


### 3.3.3 Repeatability and compatibility

The repeatability and compatibility of the analyser were evaluated through the running of a TT every 7 hours during the full drying period using $O_{2,NC}$ values, the results of which are presented in Fig. 8 and Table 3. For O$_2$ the WMO repeatability goal is ± 1 per meg (with an extended goal of ± 5 per meg) and the compatibility goal is ± 2 per meg (with an extended goal

of ± 10 per meg) (indicated by the grey dashed lines in Fig. 8) (Crotwell et al., 2019).

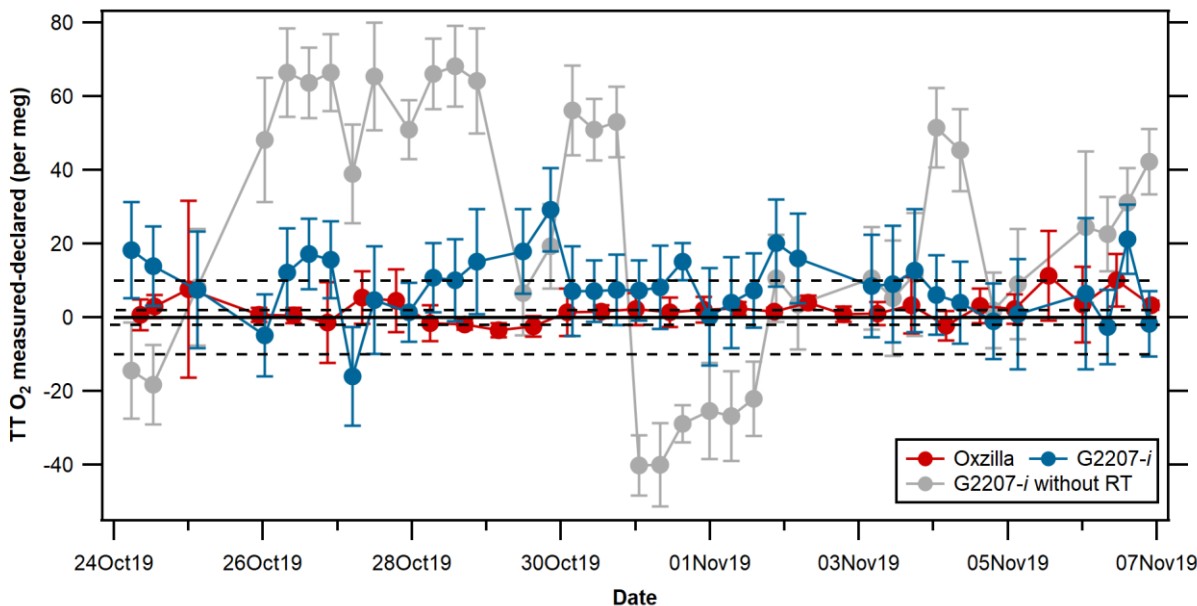

**Figure 8. TT differences from declared values (measured - declared) (± 1σ standard deviation) for the Oxzilla (red), G2207-*i* $O_{2,NC}$ (blue), and G2207-*i* $O_{2,NC}$ without RT (grey). The solid line indicated zero difference from the declared O$_2$ value of the TT, and the dashed lined indicate the WMO compatibility goal of ± 2 per meg and the extended goal of ± 10 per meg.**

**Table 3. Repeatability and compatibility goals, and achievements for each analyser**

|  | Repeatability (per meg)[a] | Compatibility (per meg)[b] |
|---|---|---|
| WMO compatibility goal | ± 1 (± 5)[c] | ± 2 (± 10)[c] |
| Oxzilla | ± 2.21 ± 1.96 | ± 3.03 ± 2.59 |
| G2207-*i* $O_{2,NC}$ without RT interpolation | ± 11.86 ± 13.83 | ± 22.88 ± 34.11 |
| G2207-*i* $O_{2,NC}$ | ± 5.69 ± 5.61 | ± 9.97 ± 6.71 |





[a] Values are calculated using the method in Kozlova and Manning (2009) and Pickers et al. (2017). Mean ± 1σ standard deviations of the average of two consecutive measurements of the TT, determined from 30 TT measurements for the Oxzilla and 37 TT measurements for the G2207-*i*, where one run is the average of 12 minutes of data. Uncertainties are given on these mean standard deviations, illustrating that the analytical repeatability is variable over time.

[b] Mean differences between the measured TT $O_2/N_2$ ratio, and the declared values determined on the VUV analyser against primary calibration standards on the SIO $O_2$ scale.

[c] WMO repeatability and compatibility goals, where the repeatability of a measurement should be at most half of the value of the compatibility goal. For $O_2$, the WMO the goals are very ambitious and not currently achievable by the $O_2$ measurement community; hence the "extended" $O_2$ goals, which are suitable for some $O_2$ applications, shown in parenthesis.


The repeatability is determined from the mean ± 1σ standard deviations of the average of two consecutive measurements of the TT. For the G2207-*i* this is equal to ± 5.69 ± 5.61 per meg, compared to ± 2.21 ± 1.96 per meg on the Oxzilla. Previous to applying the RT interpolation to the G2207-*i* data, the repeatability of the G2207-*i* was ± 11.86 ± 13.83 per meg, twice as bad as after the RT application; this is because after the RT interpolation was applied the large jumps in the TT value after a

WSS calibration were removed. In the context of the WMO repeatability goals, neither the Oxzilla nor the G2207-*i* meet the goal of ± 1 per meg. For $O_2$, the WMO the goals are very ambitious and not currently achievable by the $O_2$ measurement community; hence, the "extended" $O_2$ repeatability goal of ± 5 per meg (Crotwell et al., 2019). The Oxzilla TT results lie within this extended goal, however the G2207-*i* does not, even after the application of the RT, meaning that the G2207-*i* is not considered precise enough within the WMO goals, although it is close.

The compatibility of the analyser, which provides a measure of the compatibility to the SIO $O_2$ scale over time, and is here used as a proxy for accuracy, is determined by calculating the mean difference between the TT $O_2$ as measured by the G2207-*i* and the VUV declared value (-718 per meg). The mean absolute difference from the declared value on the VUV for the Oxzilla is 3.03 ± 2.59 per meg, this is well within the extended WMO compatibility goal of ± 10 per meg and is quite close to more stringent goal of ± 2 per meg. The compatibility of the G2207-*i* prior to the application of the RT is 22.88 ±

34.11 per meg, which is far greater than even the extended compatibility goal of ± 10 per meg. After the application of the RT interpolation the compatibility of the G2207-*i* $O_{2,NC}$ was calculated as 9.97 ± 6.71 per meg, although this is not within the WMO compatibility goal, it is just within the extended goal, which is deemed suitable for some applications in specific circumstances, such as where the signals are very large as such that reduced repeatability and compatibility does not preclude useful information from the measurements.

The compatibility and repeatability of the G2207-*i* measurements were vastly improved after the application of a 5 hourly RT, however if ignoring the TT results immediately after a new WSS calibration (i.e., after the large jumps when the RT was not applied) the repeatability without the RT interpolation is 5.21 ± 4.50 per meg, improving to 4.27 ± 4.61 per meg when the RT is applied. This is because the RT corrected for baseline drift between WSS calibrations, as the $O_2$ value of the WSSes was defined as a difference from the RT, but it does not correct for drift within the calibration period. However, as





the TT results are imprecise (as illustrated by the large error bars in Fig. 8) even if any baseline drift within a calibration period were corrected for there would likely be little improvement in the final TT results as the noise in the RT corrected TT values is primarily caused by imprecision rather than baseline drift.

**3.5 Applications of the G2207-*i* $O_2$ measurements in the calculation of fossil fuel $CO_2$**

In order to further assess the G2207-*i*'s performance in real-world applications the fully dried, RT corrected, $O_{2,NC}$
observations from WAO were used to isolate the fossil fuel component of the concurrent $CO_2$ observations and then compared to the ffCO$_2$ values calculated from APO derived from the Oxzilla $O_2$ observations following the methodology outlined in Pickers et al. (2022). The resultant ffCO$_2$ values calculated from each analyser are displayed in Fig. 9, negative ffCO$_2$ values occur when the $O_2$ observations are above (more positive) than the calculated baseline.

The measurement uncertainty was calculated as the average hourly SD on 30 October 2019, this date was chosen as it was a
particularly stable period with little variation in the TT results for both analysers (Fig. 8); the resultant uncertainty for the G2207-*i* is ± 11.19 per meg compared to ± 4.86 per meg for the Oxzilla. The uncertainty in the baseline (± 28 %), and the emission ratio uncertainty (± 22 %) are significantly larger than these measurement uncertainties (Pickers et al., 2022), but as these are the same for both analysers the additional measurement uncertainty for the G2207-*i* caused by analyser noise increases the uncertainty of the calculated ffCO$_2$ values. The average final calculated uncertainty on the ffCO$_2$ values
calculated from the Oxzilla measurements is 5.82 ppm, compared to 12.97 ppm on the G2207-*i* .

The average ffCO$_2$ value over the entire full drying period for the Oxzilla is 5.06 ppm, compared to 7.86 ppm on the G2207-*i;* the calculated ffCO$_2$ from the G2207-*i* is higher than that of the Oxzilla 73 % of the time. This difference is predominantly due to the higher $O_2$ values reported by the G2207-*i* as discussed in section 3.3.2; some of this difference also comes from the jumps in the G2207-*i* $O_2$ values which mean that the calculated baselines used for each analyser follow different trends
(Fig. A1). For example, on the 27 October 2019 and 30 October 2019 the largest difference between the calculated ffCO$_2$ values is observed (Fig. 9), on both of these dates there is a large jump in $O_2$ values from the previous day measured by the G2207-*i* following a WSS calibration (Fig. 7). Although the $O_2$ difference between the two analysers on these days are low, there was a large difference the preceding day, the days with the larger difference (due to a higher $O_2$ value reported by the G2207-*i*) in observed values pull the baseline to become more positive, thus making the difference between the ffCO$_2$
calculated from the two analysers larger on days where the observed $O_2$ difference is smaller.

Although the G2207-*i* calculated ffCO$_2$ values are often higher than those from the Oxzilla, it still follows the same trend (with some jumps in the G2207-*i* values) however, the maximum and minimum values occur at different times. The differences in ffCO$_2$ calculated from the G2207-*i* and the Oxzilla will become problematic if using the G2207-*i* analyser for top-down ffCO$_2$ quantification on an hourly basis.

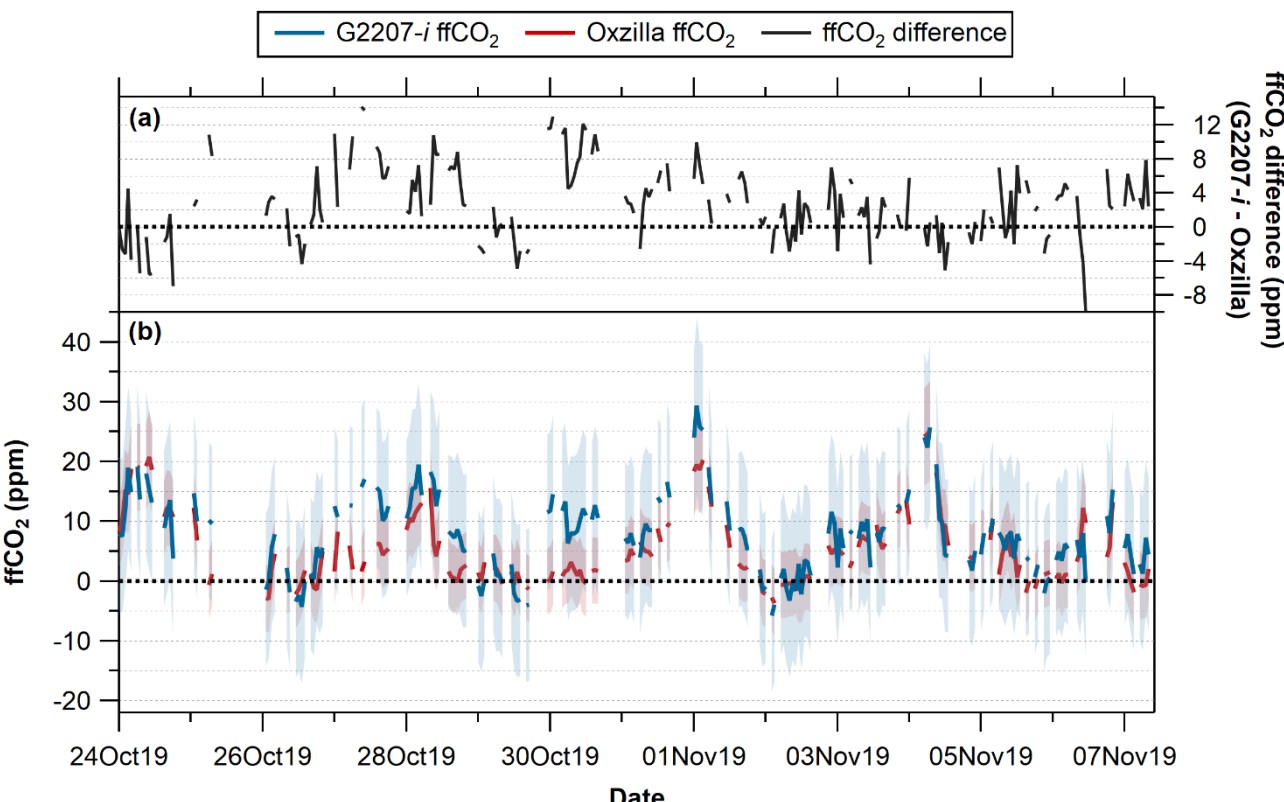

**Figure 9. (a) Calculated ffCO₂ from the G2207-*i* and the Oxzilla, shaded areas indicate uncertainty of the calculated ffCO₂. (b) Difference between the ffCO₂ calculated using the G2207-*i* and the Oxzilla O₂ (G2207-*i* – Oxzilla). Dashed black lines indicates 0 ppm. N.B. Gaps are due to threshold requirement of a minimum of 20 minutes of data for hourly averages.**

**Table 4. Comparison of ffCO₂ values calculated from the Oxzilla and G2207-*i* O₂ measurements. Average values given ±1σ standard deviation.**

|         | Oxzilla ffCO₂ (ppm) | G2207-*i* ffCO₂ (ppm) |
|---------|---------------------|-----------------------|
| Average | 5.06 ± 5.87         | 7.86 ± 6.63           |
| Maximum | 25.21               | 29.40                 |
| Minimum | -3.71               | -6.47                 |

## 4. Conclusions

The performance of the Picarro G2207-*i* under both laboratory and field conditions has been thoroughly evaluated. When running a cylinder on the G2207-*i* over 24 hours in the laboratory, we observed a large amount of noise in the raw 1 second

data, resulting in a large standard deviation in averaged data. This standard deviation is reduced over longer averaging times. During the laboratory measurement of cylinder gases with declared O₂ values, the G2207-*i* performed within the WMO





extended compatibility goal of $\pm$ 10 per meg when measuring cylinders with a negative $O_2$ per meg value. When measuring cylinders with a positive $O_2$ value, the precision and accuracy of the result worsened, thus the G2207-*i* is not recommended for use in this range.

When sampling ambient air, we found that the G2207-*i*'s built-in water correction does not, at present, sufficiently correct for the influence of water vapour even when the sample air is partially dried, and therefore recommend full drying (<1 ppm $H_2O$) of air samples. When sampling fully dried air, large step-changes in the reported $O_2$ values from the G2207-*i* were observed after each WSS calibration; the addition of a RT every 5-hours vastly reduced these jumps however they were still observable. When the RT routine was applied the repeatability of the G2207-*i* was $\pm$ 5.69 $\pm$ 5.61 per meg, falling just outside

of the WMO extended goal of $\pm$ 5 per meg, it is possible that with a more frequent RT routine this repeatability will improve. The compatibility was $\pm$ 9.97 $\pm$ 6.71 per meg, falling within the WMO extended compatibility goal for $O_2$ of $\pm$ 10 per meg. In the future, investigation into increased frequency of the running of a RT to reduce jumps in the observed $O_2$ values after a WSS calibration may improve both the repeatability and compatibility of the analyser. A key benefit of CRDS analysers is that they do not require drying of the air sample and consume considerably less gas than current methods, however, this is

not currently the case with the G2207-*i* for $O_2$ measurements.

*Data Availability.*

The G2207-*i* data from WAO and CRAM lab tests are available at https://doi.org/10.5281/zenodo.6802657 (Fleming et al., 2022)

The WAO $O_2$ and $CO_2$ in situ datasets are available from the CEDA data archives, which can be found at: https://catalogue.ceda.ac.uk/uuid/36517548500e1e4e85c97d99457e268a (Weybourne Atmospheric Observatory et al., 2006)

*Author contributions.*

LSF, ACM, and PAP developed the measurement methodology, which were conducted by LSF at UEA and WAO. AJE developed the software used to run the analyser. Investigation and visualisation were completed by LSF. Writing was

completed by LSF. Review and editing were completed by LSF, ACM, PAP and GLF.

*Competing interests.*

The authors declare that they have no conflict of interest.

*Acknowledgements.*

We are very grateful to M. Hewitt, N. Griffin, and D. Blomfield (UEA) for supporting the WAO measurements. We are also

grateful to G Lucic and M Hoffmann at Picarro Inc. for the loaning of the G2207-*i* analyser and their feedback on the manuscript. LSF is supported by the UK Natural Environment Research Council (NERC) "EnvEast" Doctoral Training Partnership, grant number NE/L002582/1. The WAO atmospheric $O_2$ and $CO_2$ measurements are supported by the Atmospheric Measurement and Observation Facility (AMOF) of the National Centre for Atmospheric Science (NCAS), in addition to NERC research grants NE/R011532/1, NE/N016238/1, NE/S004521/1.




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
