# Peer review of "Evaluating the performance of a Picarro G2207-*i* analyser for highprecision atmospheric O2 measurements"

_Atmospheric Measurement Techniques, 2022_

## Referee Comment (RC1)

**Review's comments**

**Manuscript Number: amt-2022-207**

**Title:** Evaluating the performance of a Picarro G2207-I analyzer for high-precision atmospheric O2 measurements

Authors: L. S. Fleming, A. C. Manning, P. A. Pickers, G. L. Forster, and A. J. Etchells

In this study, Fleming et al. investigated the performance of a new cavity ring down spectroscopy analyzer developed for high-precision atmospheric O2 measurements. The atmospheric O2 measurements, combined with the CO2 measurements, could potentially give us useful information about the origin of the CO2 change because most of CO2 sources have the specific O2:C exchange ratios except air-sea gas exchange. Nevertheless, the atmospheric  $O_2$  measurements are still challenging compared with the  $CO_2$ measurements. Recently developed CRDS O2 analyzer (Picarro G2207-i) nominally require no continuous working reference gas and no water vapor trap for the precise atmospheric O2 measurements. Such analyzer could enhance the possibility to extend the atmospheric O2 observation network. The authors carefully evaluated the precision, stability, influence of water vapor, and so on of the G2207-i O2 analyzer through the tank air and ambient air measurements and found that the evaluated repeatability and compatibility didn't reach the levels required for the atmospheric measurements. Although the resulting performance was regrettable, such information is quite useful to researchers in the field of the atmospheric studies and to improve the CRDS analyzer. I found that the paper is well written, well organized and contains material that should be published in AMT. I recommend this paper for publication with the following technical corrections.

Minor comments:

Page 2, line 32: "Tohjima et al., 2005a" should be "Tohjima et al. 2005".

Page 2, line 32-33: I believe that the authors well understand it that the mole fraction of a major atmospheric component, like  $O_2$ , is affected by changes in the abundance of not only trace species but also the major species because of the dilution effect. So, I think that it would be better to emphasize the above point to explain why the  $O_2/N_2$  ratio is used to

express the concentration change instead of the mole fraction.

Page 2, line 46: "(e.g. Pickers et al., 2017; Resplendy et al., 2019; ...)"?

Page 2, line 49-50: The average OR of fuel types are summarized in Keeling (1988a) not Keeling (1988b).

Page 4, line 109: What are the precision and accuracy for the water vapour mole fraction measured by G2207-i? I think such information is crucially important to evaluate the precision of the dry  $O_2$  value ( $O_{2,WC}$ ) after water vapour correction.

Page 6, line153: "((52°75'...)"?

Page 7, line 174-175: It's a just idea that how about giving the extended expression of Eq. (4) including H2O mole fraction: that is

$$\delta(O_2/N_2) = \frac{\delta O_2}{S_{O2} \cdot (1 - S_{O2})} + \frac{(CO_2 - 363.29)}{(1 - S_{O2})} + \frac{H_2O}{(1 - S_{O2})}$$

From above equation, we can easily obtain the dilution effect. Additionally, the equation is probably helpful to understand the temporal variations in the  $O_{2,NC}$  by G2207-i associated with the H2O variations shown in Figure 4 and 5 and correlation plots in Figure 6.

Page 9, Section 2.5: Eq. (4) and (5) should read as Eq. (5) and (6) because Eq. (4) already appears in Page 6 (line 148).

Page 9, line 233: "Tohjima et al, 2005b" should be "Tohjima et al., 2005".

Page 10, line 265: What's the unit of the slope of -4.26 x 10-6? Is it ppm/s?

Page 11, Figure 3: Is the coefficient of determination,  $R^2=5.37 \times 10^5$ , correct? I think it may be  $R^2=5.37 \times 10^{-5}$ .

Page 11, line 272-274: Some figures of the chemical formula are not subscripted.

Page 12, Table 2: Please check the average values of the cylinder #4 in the 8th column, the cylinder #5 in the 5th and 8th columns and the cylinder #6 in the 5th and 8th columns.

Page 12, line 289: The value of 36.0 is not standard deviation but average value in Table 2. Please check it.

Page 14, line 321: Is "(Fig. 6c and d)" correct?

Page 15, line 336: Is "(Fig. 6a and b)" correct?

Page 17, line 375: The color of the dashed lines in Figure 8 seems to be black.

Page 17, Figure 8, caption: "The solid line indicates zero ... and the dashed lines indicate the WMO..."

Page 24, line 576-581: Tohjima et al., 2005a and 2005b are same.

---

## Author Comment (AC1)

In this study, Fleming et al. investigated the performance of a new cavity ring down spectroscopy analyzer developed for high-precision atmospheric $O_2$ measurements. The atmospheric $O_2$ measurements, combined with the $CO_2$ measurements, could potentially give us useful information about the origin of the $CO_2$ change because most of $CO_2$ sources have the specific $O_2$:C exchange ratios except air-sea gas exchange. Nevertheless, the atmospheric $O_2$ measurements are still challenging compared with the $CO_2$ measurements. Recently developed CRDS $O_2$ analyzer (Picarro G2207-i) nominally require no continuous working reference gas and no water vapor trap for the precise atmospheric $O_2$ measurements. Such analyzer could enhance the possibility to extend the atmospheric $O_2$ observation network. The authors carefully evaluated the precision, stability, influence of water vapor, and so on of the G2207-i $O_2$ analyzer through the tank air and ambient air measurements and found that the evaluated repeatability and compatibility didn't reach the levels required for the atmospheric measurements. Although the resulting performance was regrettable, such information is quite useful to researchers in the field of the atmospheric studies and to improve the CRDS analyzer. I found that the paper is well written, well organized and contains material that should be published in AMT. I recommend this paper for publication with the following technical corrections.

We thank the referee very much for their positive review of our manuscript. We have addressed each of their comments below, with their comments shown in black and our responses in blue.

Minor comments:

Page 2, line 32: "Tohjima et al., 2005a" should be "Tohjima et al. 2005".

We decided to delete this citation, and to change the value from 20.94 % to 20.9 % (the word "approximately" was already in the text, so it is quite appropriate to do this).  We made this decision because since that 2005 paper was published, the background atmospheric $O_2$ mole fraction has continued to decrease, and is now closer to 20.93 %. Whereas the $O_2$ scientific community continues to use 20.94 % as part of the formal definition for reporting all $O_2$ data in "per meg" units – which we clarify later in the paper, e.g. in equations 4 and 5.

Page 2, line 32-33: I believe that the authors well understand it that the mole fraction of a major atmospheric component, like $O_2$, is affected by changes in the abundance of not only trace species but also the major species because of the dilution effect. So, I think that it would be better to emphasize the above point to explain why the $O_2/N_2$ ratio is used to express the concentration change instead of the mole fraction.

This has been reworded to: "Due to this large atmospheric background, $O_2$ measurements are sensitive to variations in the mole fractions of other atmospheric species, such as carbon dioxide ($CO_2$), due to dilution effects."

Page 2, line 46: "(e.g. Pickers et al., 2017; Resplendy et al., 2019; …)"?

This has been corrected.

Page 2, line 49-50: The average OR of fuel types are summarized in Keeling (1988a) not Keeling (1988b).

The reference we used for the average OR of fuel types is already Keeling (1988a).

Page 4, line 109: What are the precision and accuracy for the water vapour mole fraction measured by G2207-i? I think such information is crucially important to evaluate the precision of the dry $O_2$ value ($O_{2,WC}$) after water vapour correction.

Picarro datasheet information for the $H_2O$ measurement precision has been added: "The G2207-$i$ datasheet states a measurement precision of 5 ppm + 0.1 % of reading (1-σ, 5 sec) for the water vapour mole fraction."

Page 6, line153: "((52°75'…)"?

Extra open bracket removed.

Page 7, line 174-175: It's a just idea that how about giving the extended expression of Eq. (4) including $H_2O$ mole fraction: that is

$\delta(O_2/N_2)=\delta O_2/(S_{O2}\times(1-S_{O2})) + (CO_2-363.29)/(1-S_{O2}) + H_2O/(1-S_{O2})$.

From above equation, we can easily obtain the dilution effect. Additionally, the equation is probably helpful to understand the temporal variations in the $O_{2,NC}$ by G2207-i associated with the $H_2O$ variations shown in Figure 4 and 5 and correlation plots in Figure 6.

This is an interesting suggestion from the referee. But we have chosen to leave Eq. (4) as it was for the following reason: the water correction needed for the Picarro analyser includes a significant spectroscopic interference effect, not only the dilution correction; in fact problems with this spectroscopic correction is something we discuss in detail in the paper. So if we included a dilution-only correction in Eq. (4), it could mislead the reader.

Page 9, Section 2.5: Eq. (4) and (5) should read as Eq. (5) and (6) because Eq. (4) already appears in Page 6 (line 148).

This has now been corrected.

Page 9, line 233: "Tohjima et al, 2005b" should be "Tohjima et al., 2005".

This reference has been corrected.

Page 10, line 265: What's the unit of the slope of -4.26 x $10^{-6}$? Is it ppm/s?

Yes, the units have now been added.

Page 11, Figure 3: Is the coefficient of determination, $R^2=5.37\times10^5$, correct? I think it may be $R^2=5.37\times10^{-5}$.

Yes, this has now been corrected to $R^2 = 5.37\times10^{-5}$.

Page 11, line 272-274: Some figures of the chemical formula are not subscripted.

Figures have now been subscripted.

Page 12, Table 2: Please check the average values of the cylinder #4 in the 8[th] column, the cylinder #5 in the 5[th] and 8[th] columns and the cylinder #6 in the 5[th] and 8th columns.

We checked and these values are correct. Note that in columns 5 and 8, we have calculated the mean of the absolute values of the differences (as stated in the column headings). We have done this because if the first run had a difference of -20 per meg and the second run had a difference of 20 per meg, then the average difference would be 0 per meg, which would be very misleading in terms of the imprecision of the analyses.

Page 12, line 289: The value of 36.0 is not standard deviation but average value in Table 2. Please check it.

This value has been corrected to 19.5 per meg, as per Table 2.

Page 14, line 321: Is "(Fig. 6c and d)" correct?

This has been corrected to "(Fig. 6a and b)".

Page 15, line 336: Is "(Fig. 6a and b)" correct?

This has been corrected to "(Fig. 6c and d)".

Page 17, line 375: The color of the dashed lines in Figure 8 seems to be black.

Line colour has been removed.

Page 17, Figure 8, caption: "The solid line indicates zero … and the dashed lines indicate the WMO…"

This correction has been made.

Page 24, line 576-581: Tohjima et al., 2005a and 2005b are same.

The duplication of this reference has been fixed.

---

## Author Comment (AC2)

We thank the referee very much for their positive review of our manuscript. We have addressed each of their comments below, with their comments shown in black and our responses in blue.

Fleming and coauthors present an evaluation of a commercial cavity ring-down spectrometer, both in terms of its ability to meet WMO compatibility criteria for O2/N2 under specific conditions, and its suitability for in situ measurements.

The suggested advantages of the Picarro analyzer are that it can be run without sample drying and does not require continuous reference gas flow. This would make it attractive for installation at a remote site. However, the authors show that the instrument is unsuitable for such an application, due to the large artifacts the analyzer is subject to under such conditions. As the authors show, it does perform reasonably well when measuring tanks. For this reason I think the authors focus a bit too much on the compatibility/repeatibility goals of the Picarro under laboratory conditions. Much more telling is the in situ data. I am not sure the ffCO2 discussion adds much to the paper, since it relies on CO2 measurements not made by the Picarro analyzer. The in situ comparison presented is more to the point, and sufficient for the demonstration. The authors could even cut the ffCO2 comparison from the paper to reduce the length, in my opinion.

We carefully considered this suggestion to delete the $ffCO_2$ comparison text (section 3.5), but decided against it. In our opinion, this section provides a valuable case study of an application of $O_2$ measurements that is becoming increasingly used by scientists and which has significant policy relevance (see, for example, Pickers *et al.*, Science Advances, 2022 (https://www.science.org/doi/10.1126/sciadv.abl9250)). Furthermore, this section demonstrates that for an application where one does not need to meet the WMO compatibility goal in order to make meaningful conclusions, the Picarro $O_2$ analyser, in its current form, is still not good enough.

The point about this section relying on $CO_2$ measurements not made by the Picarro analyser is not relevant. Every atmospheric $O_2$ application also requires $CO_2$ measurements to be made, but no analyser exists that measures both $O_2$ and $CO_2$. So every laboratory or field station making $O_2$ measurements also needs an independent $CO_2$ analyser making concurrent measurements. Also, the measurement uncertainty of our $CO_2$ analyser is at least an order of magnitude lower than that of the Picarro $O_2$ analyser, thus the uncertainty we report for the $ffCO_2$ calculations in this section are almost entirely based on the Picarro $O_2$ measurement uncertainty (as well as other uncertainties inherent in the $ffCO_2$ methodology as we discuss, for example, uncertainty in the $O_2$ baseline determination).

I think this is an excellent paper and of interest to the AMT readership. I recommend publication with only minor comments.

Minor Comments

L17 and throughout: It would be easier on the reader to stick with a single unit, rather than switching between ppm and per meg.

We agree that using per meg consistently would be less complicated, however the cylinder data used to calculate the Allan deviation was not calibrated. But to assist the reader, approximate per meg values relating to these ppm values using a conversion factor of 4.8 per meg/ppm have now been added in parenthesis.

L17: I think the wording needs refining here, do you mean that the highest precision possible was found at 300 seconds? What does it mean to report an Allan deviation as 1 standard deviation? For an abstract I think it's sufficient to say that you estimated the precision to be 1 ppm, reported as 1 sigma, from 300 second means.

This has been rewritten and now relates to the best precision achievable: "we found that the best precision was achieved with 30 minute averaging and was ± 0.5 ppm (~ ± 2.4 per meg)" - With the updated ppm values being pulled from the rewritten discussion of the updated Figure 2 (L264).

L21: pre-dried is confusing, suggest "dried". The grammar is a little off in this sentence due to the mixing of tenses.

The sentence has been changed to have a consistent tense, and we removed "pre-" from "pre-dried". It now reads: "When sample air was dried and a 5-hourly baseline correction with a reference gas cylinder was employed,…"

L24: The abstract is quite long, suggest cutting "(sometimes known as a "surveillance tank")"

(sometimes known as a "surveillance tank") has been removed from the abstract, and added into the main body (p8 L212).

L43: Better to give the increase of CO2 over the same period as O2 (past three decades).

This has been rewritten to: "over the same period, atmospheric $CO_2$ has been increasing at an average rate of 2 ppm $yr^{-1}$"

L55: I think it's confusing to give an approximate definition for APO when saying it is defined as, better to give the actual equation.

The full equation is likely to confuse readers as it includes (very minor) $CH_4$ and CO terms, but we also understand the referee's point that the "approximate" sign is confusing. So rather than write the full equation, we have changed the approximate sign to an equals sign, and added to the text: "… and where we have ignored very minor influences from methane and carbon monoxide." A minus sign was also added to the $O_2$:$CO_2$ OR value of -1.1, as it had been inadvertently missed out.

L62: It would be good to define compatibility here...in L66 it seems conflated with precision. Doesn't compatibility combine accuracy and precision into a single metric?

Compatibility has now been defined on L67-68, with the addition of the following: "where compatibility refers to the acceptable level of agreement between two field stations or laboratories when measuring the same air sample." The term precision here has been replaced with repeatability, as is used throughout the rest of the paper, and has also been defined here for clarification (L73-74) as: "Repeatability refers to the closeness of agreement between results of measurements of the same measure (which is also sometimes referred to as the measurement system's precision)".

L81-84: To measure O2 with high precision and accuracy you need all of these things. The author is suggesting that they can all be contained within a single box, which is certainlt convenient. "Revolutionize" seems a bit too strong to me. There might be some savings in avoiding the continuous use of a reference gas, but the Picarro analyzer is expensive, and all of the other expensive, labor intensive aspects to making in situ measurements would still be needed.

The word "revolutionise" has been changed to the softer "advance".

However, we don't completely agree with the referee – the potential of the Picarro $O_2$ analyser is much more than simply less use of reference gases; furthermore, we disagree that the Picarro analyser is any more expensive than other $O_2$ systems, especially when taking into account the significant labour costs needed to improve other systems to sufficient precision and accuracy.

Thus, we believe that if the Picarro analyser were to work as it was intended, then it would have a very significant impact on the field of high-precision $O_2$ measurements, primarily by making such technically difficult and challenging measurements much more accessible to a wider scientific community via an easier to use "off-the-shelf" analyser.

L89-91: And yet the authors go on to show that the instrument does NOT have all of these advantages. I think this needs some rephrasing..."the vendor suggests that" or "it is intended for" use without drying, cal gas, etc. To be fair to Picarro, maybe this is not what they had in mind. There are other applications for this instrument beyond the small field of high-precision atmospheric monitoring.

This sentence has been rephrased to: ", it is intended that the G2207-*i* should not require a continuous reference gas supply".

L125: What's the flow rate, and how big is the cell? Does it really take 8 minutes to flush it? This is an extremely long e-folding time. It would be nice to see some of the actual calibration data. If the sample air is wet and the calibration gases are dry, isn't it more likely it's a surface effect rather than a purging issue?

The referee is correct: flushing does not take 8 minutes, and for the CRAM Lab tests we were switching only between dry cylinders (that is, there were no issues of changing between wet and dry air, and thus no surface effects to be concerned with). 20 and 8 minutes (analysis and flushing times, respectively) were chosen to match the times during subsequent field tests at our WAO station. The following phrase has been added to reflect this (L137: "… and to maintain consistency with the flushing time employed in subsequent WAO tests (section 2.3.2)". Additional explanation of the cylinder run-time has also been added to section 2.3.2 (L216-217).

Figure 1: How is pressure/flow control maintained for the Oxzilla? I see no pump depicted.

The Oxzilla and Siemens have an independent flow and pressure control set-up (which includes a pump), sampling from a different AAI to the Oxzilla, which contains too many components and is unnecessary to be depicted in this gas handling diagram. So for clarity L179 has been edited to read: "A diagram of the gas handling set-up for the G2207-*i* at WAO is displayed in Fig.1." The caption for Figure 1 has also been edited to "Calibration gases were shared with the established $O_2$ and $CO_2$ system (using V4), but the established system has its own AAI, pump, drying system, and pressure and flow control (not depicted here)."

L173: change "scales," to "scales. This"

Changed.

L177: There are also surface effects to consider, the dilution effect is not the sole reason.

Yes, good point. But in the case of $O_2$ (much less true for $CO_2$ and other trace gases), the dilution effect far outweighs any possible biases from surface effects. Therefore we chose not to change this text.

L202: Really? Again, I find this surprising.

The referee is correct in that the 8 minutes isn't necessary solely for flushing of the Picarro's cell. But, particularly for $O_2$ measurement, we find that relatively long flushing times are needed whenever valves are cycled and a new air stream is introduced. We (the high-precision atmospheric $O_2$ community) suspect this is related to equilibration times needed on the surfaces of all wetted materials for all components (tubing, valves, pressure regulators, etc). Thus an 8-minute flushing time was chosen to match the existing $O_2$ measurement system, as this has been proven to be sufficient. The following sentence has been added to clarify this: "A flushing period of 8 minutes and averaging time of 12 minutes were chosen to match that of the established system."

L250: Please give +/- on cavity pressure, temperature, and flow.

According to the Picarro data files the cavity pressure's standard deviation was ± 0.00146 torr and the cell temperature's standard deviation was ± 0.000306 °C. We don't think the sensors are precise enough to report standard deviations to this level, so they are effectively zero.

Figure 2: It would be nice to see the x-axis extended here, since the time horizon for the RT (5 hours) is outside of the plot.

When the x-axis is increased to 5-hours, the noise overtakes the signal and the important features in the plot can no longer be seen. We also don't think that the Allan deviation directly relates to the RT interval, but to the measurement averaging time. We have extended the x-axis so that 1 hour is visible, as this is the frequency we are averaging when investigating the in-situ air measurements. The text has also been amended to reflect this (L265): "Precision then continues to improve until around a 30 minute averaging time where a precision of ~0.5 ppm (~2.4 per meg) is reached, and remains around that value for averaging times up to around 1 hour"

L366: I don't fully understand this. It looks like the grey points are jumping at calibration intervals, and the calibration coefficients were not interpolated between calibrations, but applied stepwise. Why would the Picarro instrument's baseline jump always at calibration times? This is actually the only shortcoming of the paper--it would be nice if the author's could speculate more as to what is going on to cause these baseline shifts.

The calibration coefficients were applied step-wise, which is why there is a step change after each calibration. The baseline doesn't jump at calibration times, but it is drifting- hence when a new calibration is applied the baseline shift is applied to the measurements. The reference tank correction is interpolated, which corrects for the baseline drift.

For clarity the text has been edited to read: "the large jumps in the G2207-i $O_{2,NC}$ values following WSS calibrations (see Fig. 7b, grey points) are caused by a drift in the analyser's baseline, which only become applied to the data after each calibration. These jumps were reduced through the application of the 5-hour RT interpolation procedure (see 7b, blue points) which constrained the baseline drift (refer to Section 2.3.2). After the application of the RT interpolation the jumps between WSS calibrations were vastly reduced (see Fig. 7), thus the ffCO2 results in section 3.5 have this correction applied."

Figure 8: I think it would be better to drop the no RT Picarro data here, zoom in on the y-axis, and make the points open circles (and smaller)--it is hard to see the data which matters, which is the Oxzilla vs the RT-corrected Picarro.

The no RT data has been removed from the figure, but retained in Table 3 and the discussion in the text.

L400: "which provides a measure of the compatibility to the SIO O2 scale over time" -- I'm not sure that's quite correct, unless the tanks are being remeasured at SIO for each comparison.

This sentence has been removed.

L455: Maybe it's worth pointing out here (or earlier) that for in situ measurements, the O2/N2 ratio will be changing over tens of minutes. Averaging down pure random noise is not the same as averaging observations over an hour.

The following sentence has been added to section 3.1 (L266) : "It should be noted, that unlike the hourly average and standard deviation obtained from measurement of cylinder air, the hourly averages of atmospheric data also contain natural variability in addition to analyser-related noise and drift."